# Preparation of cyanobacteria-enhanced poly (vinyl)alcohol-based films with resistance to blue-violet light / red light and water

Sainan Chen[1]☯*, Di Liu[2]☯, Min Qian[2]☯, Li Xu[2‡], Ying Li[2‡], Haozhong Sun[2‡], Xi Wang[2‡], Haiyun Zhou[1‡], Jian Bao[1‡], Changyan Xu[2‡]*

1 Jiangsu Provincial Key Laboratory of Environmental Engineering, Jiangsu Academy of Environmental Science, Nanjing, Jiangsu Province, China, 2 College of Materials Science and Engineering, Nanjing Forestry University, Nanjing, Jiangsu Province, China

☯ These authors contributed equally to this work.
‡ These authors also contributed equally to this work.
* changyanxu1999@163.com (CX); csn@jshb.gov.cn (SC)

**Data Availability Statement:** All relevant data are within the manuscript and its Supporting Information files.

## Abstract

The harmful cyanobacteria blooms which usually form in spring and summer, cause global eutrophication of freshwater and coastal marine ecosystems. This study tried to utilize cyanobacteria as a raw material to produce biological poly(vinyl)alcohol-based films. Cyanobacteria was firstly modified with poly(ethylene glycol), guanidine hydrochloride, carboxymethyl cellulose and 3-glycidoxypropyltrimethoxysilane as plasticizer, modifier, toughening agent and coupling agent, respectively. And then the modified cyanobacteria was introduced to poly(vinyl)alcohol and cellulose nanofibers/poly(vinyl)alcohol matrix to improve the barrier properties of poly(vinyl)alcohol to light and water. Compared with poly(vinyl)alcohol and cellulose nanofibers/poly(vinyl)alcohol films, the obtained cyanobacteria/poly(vinyl) alcohol and the cyanobacteria/cellulose nanofibers/poly(vinyl)alcohol composites exhibit better resistance to light and water. More interestingly, we found that after adding cyanobacteria, the poly(vinyl)alcohol-based films present better barrier properties to blue-violet light and red light. In adddition, introducing cyanobacteria into poly(vinyl)alcohol or cellulose nanofibers/poly(vinyl)alcohol matrix increases the surface roughness and contact angle to water of the composites.

## Introduction

The global bioplastics for packaging industry is forecast to grow from 2017–2022 at an annual average rate of 17% to a market value of almost $7.2 billion according to a report from Smithers P.[1]. As one of the important bioplastics, poly(vinyl)alcohol (PVA) has a broad spectrum of applications in fabricating coatings and films in food applications [2]. It is also a main ingredient in lubricant or temporary skin covers or wound dressings in pharmaceutical applications [2–5] because of its biocompatibility, biodegradability, non-toxicity, inexpensiveness [6], easy process situation, excellent transparency, surface activity [3], and suitable film-forming capabilities [4]. However, on the other hand, its poor barrier property especially to light or

**Funding:** This research was funded by the Postgraduate Research & Practice Innovation Program of Jiangsu Province (grant number SJKY19_0868) and Key Laboratory of Jiangsu Province Environmental Engineering for Open Project Foundation (grant number ZX2017009). Initials of author: SNC

**Competing interests:** The authors have declared that no competing interests exist.

water and thermal stability greatly limit the application development of PVA. Some studies have shown that the presence of hydrogen bond groups in PVA structure and its ability of forming hydrogen bond make it suitable for PVA to mix with other materials to improve its functional characteristics [3,4,7]. For example, PVA/chitosan blend membranes produced by Gupta, B., et al [8] and PVA/carboxymethyl cellulose composite films manufactured by Kanatt, S. R., et al [9] presented better properties than PVA in the view of applications in packaging industry. The main reason why Kanatt chose nanocellulose as the strengthening phase of PVA was considering the excellent properties and availability of nanocellulose. Cellulose nanofibers (CNF) are renewable, biocompatible and biodegradable [10,11], and can be extracted from biomass such as wood [12], bamboo [13] and crop straw [14]. It has been widely used to improve the tensile and thermal properties of PVA [15–17].

As a type of prokaryote, cyanobacteria (CY) can absorb a large amount of nitrogen and phosphorus, and then rapidly proliferates to form harmful cyanobacteria blooms in spring and summer, resulting in a global problem, eutrophication of freshwater and coastal marine eco-systems [18,19]. Prevention and control of cyanobacteria pollution has become a common concern among scientists and entrepreneurs. It is an effective way to reduce the pollution of cyanobacteria to utilize it as resources. So far, there have been two kinds of methods to utilize cyanobacteria as resources. One is engineering cyanobacteria cell factories for manufacturing various chemical compounds, such as lactic acid [20], C3 platform chemicals [21], 2, 3-butane-diol [22], fatty acid [23], isobutylaldehyde [24], n-butanol [25] and bioplastic PHAs [26,27]. However, this route cannot effectively solve the problem of cyanobacteria excessive reproduction. The other method of utilizing cyanobacteria resources is to try to directly use cyanobacteria as raw materials to produce biological composites [28], biofuel [29], biological fertilizer [30]. In addition, some researchers proposed that cyanobacteria could be used as a biological agent to improve the degradation ability of soil to various pollutants [31]. Whereas, these methods need to be explored further because they are still in trials and have not been widely reported.

According to our investigation, there had been no reports of cyanobacteria being directly used to prepare composite films. Considering the insufficient resistance to light and water of PVA as a packaging material, cyanobacteria was introduced into PVA matrix in this paper. In our early elemental analysis test, the contents of carbon, nitrogen and hydrogen in the cyanobacteria sample were 45.02%, 5.98% and 9.11%, respectively. It was found that in the preliminary experiment directly mixing cyanobacteria, PVA and CNF to prepare the PVA-based films led to a poor interface combination because of the microphase separation between cyanobacteria and PVA and CNF. In this paper, cyanobacteria/cellulose nanofibers/poly(vinyl) alcohol (CY/CNF/PVA) bioplastic films were fabricated with poly(ethylene glycol), guanidine hydrochloride, carboxymethyl cellulose and 3-glycidoxypropyltrimethoxysilane as plasticizer, modifier, toughening agent and coupling agent, respectively. The purpose of this study is to explore the possibility of direct utilizing cyanobacteria as raw materials to prepare CY/CNF/PVA films, and to improve the light and water resistance of PVA films.

## Materials and methods

### Materials

The cyanobacteria (CY) collected in this experiment were collected from the algal and water separation station in Yixing Bafang port, which is located on the western bank of Taihu lake (moderately nutrient-rich water quality), with a straight-line distance of 1.5 km from the long-depth high-speed (G25) and 0.8 km from the dubian line (S230). Due to the bloom of cyanobacteria, they can be artificial collected directly on the surface of the lake. The algae pump and

the associated diversion and containment equipment were used to collect cyanobacteria from the surface. The surrounding ecological environment was not destroyed during the process. The fresh cyanobacteria was transported to the material laboratory of nanjing forestry university on the same day, and then firstly dried by sunlight for 2 days. After that dried it for at less 48 hours at 60°C until the moisture content was less than 20%. After screening, 100 mesh cyanobacteria powder was sealed in plastic bags for later use.

This study was approved by the Cyanobacteria Office of Wuxi Water Resources Bureau. All cyanobacteria used were collected by the staff of the Cyanobacteria Office.

Polyethylene glycol (200 molecular weight) were from Yonghua Chemical Technology Co., Ltd (Jiangsu Province, China). Polyvinyl alcohol (750±50 degree of polymerization), guanidine hydrochloride, Sodium carboxymethyl cellulose and 3-(Glycidoxypropyl)triethoxysilane (97 wt%) came from Sinopsin Chemical Reagent Co., Ltd (Jiangsu Province, China). Nanocellulose (shown in Table 1) were obtained from Zhongshan NFC Bio-materials Co., Ltd,(Guangdong Province, China).

## Preparation of the PVA-based films

**Experimental scheme.** The experimental scheme and formula of the the PVA-based films are shown in Table 2. No. 1 (PVA film) and No. 7 (CNF/PVA film) is the control to No. 2-No.6 and No. 8-No.12 respectively for investigating the influence of cyanobacteria on the properties of PVA films and CNF/PVA films.

**Preparation of the films.** 2g PVA was added to 18 ml deionized water and stirred for 2 hours at 95°C to obtain 10 wt% PVA solution for later use. Meanwhile, 8g 2.5 wt% CNF solution was added to 12 ml deionized water, stirred at room temperature for 1 hour and sonicated for 40 min (XO-1200, Nanjing Xianou Instrument Manufacturing Co., Ltd., China) to obtain 1 wt% CNF solution for later use. The prepared 10 wt% PVA solution and 1 wt% CNF solution were mixed by weight ratio of 2:1 to obtain CNF/PVA solution for later use.

The PVA-based films were prepared by a solution casting method. According to the formulation in Table 2, a certain amount of 10 wt% PVA solution was poured into a petri dish, and put in an oven (101-2BS, Beijing Hengnuolixing Technology Co., Ltd., China) for drying at 40°C for 24 hours to get the PVA film (No. 1, shown in Fig 1). In the same way, the CNF/PVA film (No. 7, shown in Fig 1) was obtained.

Cyanobacteria powder (1g, absolute dry weight) was mixed with deionized water (46.75g) to obtain cyanobacteria homogenization. Poly(ethylene glycol) (1g), guanidine hydrochloride (0.75g) and carboxymethyl cellulose (0.5g) was added successively into the cyanobacteria homogenization. After stirring at room temperature for 2 hours, the modified cyanobacterial was obtained, as showm in Fig 1. According to the formulation in Table 2, a certain amount of

**Table 1. Information of the purchased nanocellulose.**

| Name | Specification |
|---|---|
| Product model | NFC1802N |
| Raw materials | Coniferous wood pulp fibre |
| Concentration | 2.5±0.5% |
| Carboxyl content | 1.02 mmol/g |
| Length | 1~2 μm |
| Diameter | 30 nm |
| Aspect ratio | 30~100 |
| Net weight | 250 g |

**Table 2. Experimental scheme and formula of the PVA-based films.**

| Film No. | CM[a] | | PVA[a] | CNF[a] | (Glycidoxypropyl)triethoxysilane[a] |
|---|---|---|---|---|---|
| 1 | 0 g | 0 wt%[b] | 20 g | 0 g | 0 g |
| 2 | 1 g | 5 wt%[b] | 20 g | 0 g | 0.5 g |
| 3 | 3 g | 15 wt%[b] | 20 g | 0 g | 1.0 g |
| 4 | 5 g | 25 wt%[b] | 20 g | 0 g | 1.5 g |
| 5 | 7 g | 35 wt%[b] | 20 g | 0 g | 2.0 g |
| 6 | 9 g | 45 wt%[b] | 20 g | 0 g | 2.5 g |
| 7 | 0 g | 0 wt%[b] | 13.3 g | 6.7 g | 0 g |
| 8 | 1 g | 5 wt%[b] | 13.3 g | 6.7 g | 0.5 g |
| 9 | 3 g | 15 wt%[b] | 13.3 g | 6.7 g | 1.0 g |
| 10 | 5 g | 25 wt%[b] | 13.3 g | 6.7 g | 1.5 g |
| 11 | 7 g | 35 wt%[b] | 13.3 g | 6.7 g | 2.0 g |
| 12 | 9 g | 45 wt%[b] | 13.3 g | 6.7 g | 2.5 g |

[a] These are the weight of solution (the modified cyanobacteria, 10 wt% PVA solution, 1 wt% CNF solution and 3-(Glycidoxypropyl)triethoxysilane solution).

[b] The weight content of CM in film No.1-6 is CM to PVA. The weight content of CM in film No.7-12 is CM to CNF/PVA.

modified cyanobacteria, referred to as CM, was mixed with PVA solution or CNF/PVA solution. After adding 3-(Glycidoxypropyl)triethoxysilane solution, the mixture was heated and stirred at 55˚C for 1 hour, and casted in a petri dish, and then oven-dried (101-2BS, Beijing Hengnuolixing Technology Co., Ltd., China) at 40˚C for 24 hours, resulting in the CY/PVA films (No.2 - No.6, shown in Fig 1) and CY/CNF/PVA films (No.8 - No.12, shown in Fig 1).

## Characteristics of the PVA-based films

**Fourier transform infrared spectroscopy (FTIR) analysis of the film samples.** FTIR spectra of the film samples (PVA, CY/PVA, CNF/PVA, CY/CNF/PVA) have been acquired

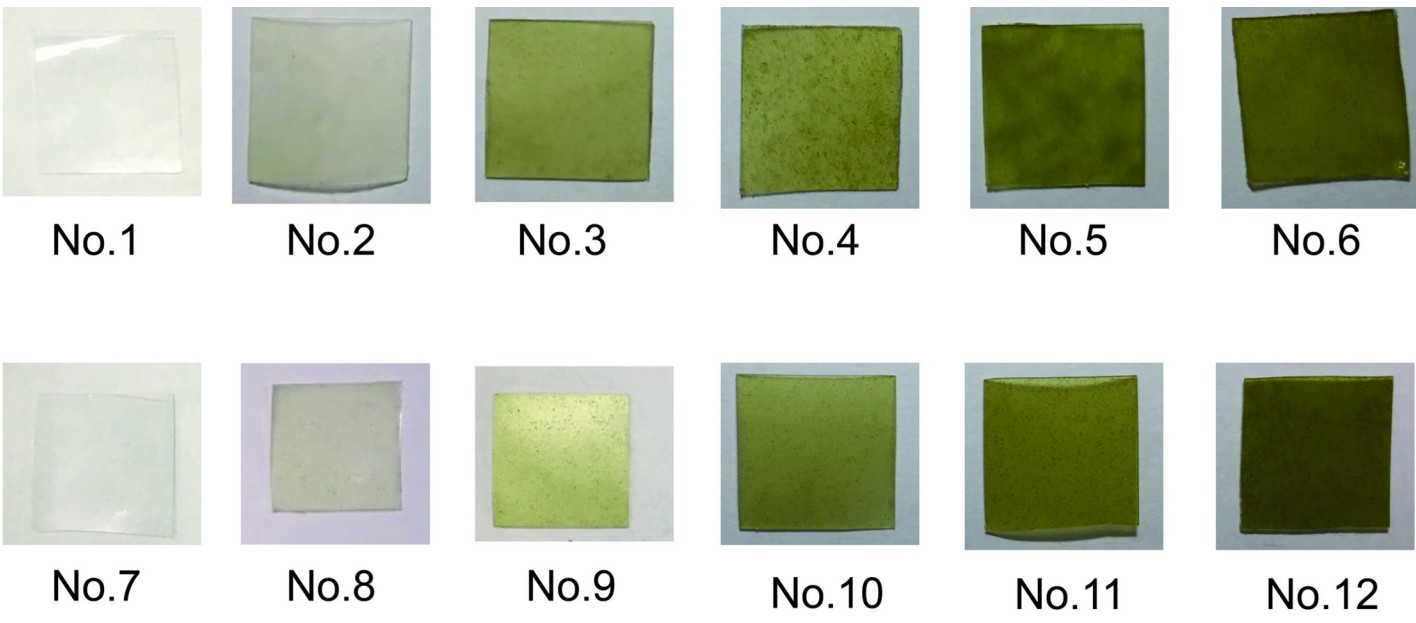

**Fig 1. Pictures of the PVA-based films.**

with a Fourier transform spectrometer (NICOLET IS10 Thermo SCIENTIFIC, Inc, USA) operating in the mode of Smart iTR diamond ATR and the range from 500 $cm^{-1}$ to 4000 $cm^{-1}$. The baseline correction was done using the accompanying built-in software.

**Morphology analysis of the PVA-based films.** The fracture morphology analysis of the PVA-based films was observed by a cold field emission scanning electron microscope (Hitachi Regulus 8200, Japan).

The section in the direction of film thickness was placed up and attached to the conductive tape on the sample stand. The gold plating on the section of the film sample was conducted under the condition of 20 mA of gold spraying current and less than 7 Pa of vacuum degree for 80~100 seconds. The working voltage of the scanning electron microscope was 5kv.

**Barrier properties of the PVA-based films to light.** A spectrophotometer U-4001 (Hewlett-Packard Co., Santa Clara, CA, USA) was used to analyze the barrier properties of the PVA-based films to ultraviolet and visible light. The transmittance of the films to light in the range of 200–800 nm was measured.

**Surface contact angle to distilled water of the PVA-based films.** After drying in an oven (Electric blast drying box, 101-2BS, Beijing Hengnuolixing Technology Co., Ltd, China) at 100˚C for 24 hours, the PVA-based film was cut to pieces with a size of 20 mm × 20 mm (length×width). The surface contact angle measurement of the sample to distilled water was carried out in an automatic single fiber contact angle measuring instrument (OCA40, Data-Physics Intruments GmbH, German).

**Water resistance of the PVA-based films.** The swelling of a sample is usually used to indicate the water resistance of a material. Operating methods refer to ASTM d570-98. The PVA-based composite film with a size of 2 cm × 2 cm was firstly weighed ($W_p$), then oven-dried at 100˚C for 24 hours and weighed ($W_i$). The degree of moisture content (X) of the composite films was calculated by Eq (1),

$$X = (W_p - W_i)/W_p \times 100\% \tag{1}$$

The sample after oven-dries was immersed in 50-ml distilled water for 24 hours at room temperature. Drying the moisture of the sample surface, the sample was weighed again ($W_f$). The degree of swelling (A) [32] of the sample was calculated by Eq (2),

$$A = [(W_f - W_i)/W_i] \times 100\% \tag{2}$$

**Oxygen permeability of the PVA-based films.** The film sample with a size of 10cm was firstly oven-dried at 100˚C for 24 hours, and then tested its oxygen permeability with a differential pressure gas permeameter (PERME VAC-V2, Jinan Labthink Electromechanical Technology Co., Ltd., China). The analysis conditions were 50% relative humidity, room temperature (23˚C) , 10% proportion mode, GTR≥1 and 12 hours.

## Results and discussions

### Moisture content and thickness of the PVA-based films

Table 3 lists the moisture and thickness of the prepared PVA, PVA/CNF, CY/PVA and CY/CNF/PVA films. The data of the moisture content and thickness of the film with each formulation are the average of the 3 replica. It can be seen that the moisture content and the thickness of the composite films vary within the range of 3–6% and 0.2–0.5 mm, respectively.

**Table 3. Moisture content and thickness of the PVA-based films.**

| Film No. | Moisture content (%) | Thickness (mm) |
|---|---|---|
| No.1 | 4.29±0.92 | 0.22±1.05 |
| No.2 | 4.24±1.17 | 0.25±0.29 |
| No.3 | 5.26±1.01 | 0.34±0.97 |
| No.4 | 5.76±1.18 | 0.37±0.33 |
| No.5 | 3.97±0.92 | 0.42±0.06 |
| No.6 | 5.07±0.64 | 0.49±1.12 |
| No.7 | 3.26±0.85 | 0.21±1.07 |
| No.8 | 5.89±1.10 | 0.28±1.02 |
| No.9 | 3.20±0.91 | 0.28±1.29 |
| No.10 | 5.03±0.12 | 0.37±1.02 |
| No.11 | 5.56±1.53 | 0.45±0.08 |
| No.12 | 3.98±0.09 | 0.48±0.74 |

## FTIR analysis of cyanobacteria and the PVA-based films

The FTIR spectra of the cyanobacteria used in this experiment, the obtained the PVA-based films are shown in Fig 2. The main characteristic peaks of PVA and PVA/CNF are consistent with those of the composites in the previous reports [33–36]. In the spectra of the PVA film and CNF/PVA composite, the peaks at 3276 cm$^{-1}$ are attributed to the stretching vibration absorption peak of hydrogen bonding group. Although there exist a large amount of hydrogen groups in the structure of PVA, CNF and cyanobacteria, all vibration peaks at 3276 cm$^{-1}$ in the spectra of the CY/CNF/PVA composites are weaker than that of PVA film or CNF/PVA composite. It may be due to the fact that the active amine in guanidine hydrochloride reacts with the inter- and intra-molecular hydrogen bonding of the lipids and starch in cyanobacteria [18].

In FTIR spectra of all samples, the absorption peaks at 2927 cm$^{-1}$, 1661 cm$^{-1}$, 1408 cm$^{-1}$ and 1200–1000 cm$^{-1}$ respectively belong to the anti-symmetric stretching vibration of hydrogen bond, the stretching vibration of carbon-oxygen double bond, the bending vibration of methylene group, and the characteristic spectral region of polysaccharide [33–36]. Each sample presents a characteristic spectral region of polysaccharide, which is attributed to the cyanobacteria

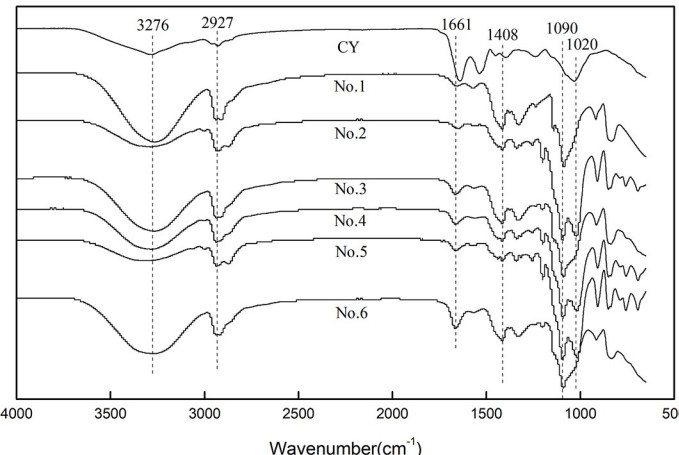
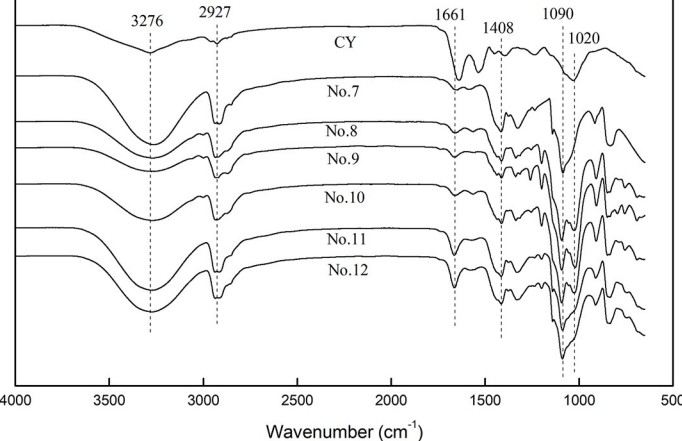

**Fig 2. FTIR spectra of the CY, PVA, PVA/CNF, CY/PVA and CY/PVA/CNF films.**

or CNF in the sample. In addition, with the increase of cyanobacteria content, the absorption peak at 1661 cm$^{-1}$ in the spectrum of the sample enhances owing to the peptide bonding of phycocyanin in cyanobacteria [37].

The peak at 1090 cm$^{-1}$ is the characteristic absorption peak of PVA, which still appears in the spectra of all CY/CNF/PVA samples. The peak at 1020 cm$^{-1}$ which is the characteristic absorption peak of cyanobacteria, still appears in the spectra of the CY/CNF/PVA samples with lower modified cyanobacteria content (No. 2, No.4, No.5, and No.8-No.10), indicating the successful blending of the modified cyanobacteria with the PVA and PVA/CNF. However, for the samples (No.3, No. 6, No.11, and No.12) with more modified cyanobacteria content, the spectra does not present the absorption peak at 1020 cm$^{-1}$, showing that too much modified cyanobacteria leads to poor mixing of modified cyanobacteria with PVA and PVA/CNF.

Besides, a series of peaks appear at 800–400 cm$^{-1}$ in the spectra of all CY/CNF/PVA samples, which are ascribed to the out-of-plane bending vibration of hydroxy groups[33–36], demonstrating that there exist some inter-molecular hydroxy groups in the obtained CY/CNF/PVA samples.

## Barrier properties of the PVA-based films to ultraviolet and visible light

The transmittance curves of the PVA-based films for ultraviolet and visible light at a wavelength of 200–800 nm are shown in Fig 3. It can be seen that introducing CNF to PVA decreases the transmittance of the PVC for ultraviolet and visible light, and further adding cyanobacteria to CNF/PVA matrix further decreases the transmittance of the CNF/PVA. The higher the cyanobacteria content, the lower the light transmittance of the CY/CNF/PVA films. It means that the barrier property of PVA to ultraviolet and visible light can be enhanced by CNF and CY. This phenomenon can be explained by the fact that the particle in CNF and cyanobacteria sizes larger than the corresponding wavelength would obstruct light[38]. In addition, all spectral lines of CY/PVA and CY/CNF/PVA samples in Fig 3 show two strong absorption peaks at 403 nm and 666 nm. Table 4 lists the transmission the of the PVA-based films at two critical wavelengths. Light with wavelength less than 400 nm is ultraviolet light, and light with wavelength between 400nm and 760nm is visible light, in which the wavelength of 620-760nm belongs to red light, and that of 400–464 nm belongs to blue violet light. Therefore, a conclusion it can be made that the prepared CY/PVA and CY/CNF/PVA films have strong barrier properties to red and blue violet light. It is due to the chlorophyll a in cyanobacteria[37], which mainly absorbs red and blue violet light [39]. This property is particularly

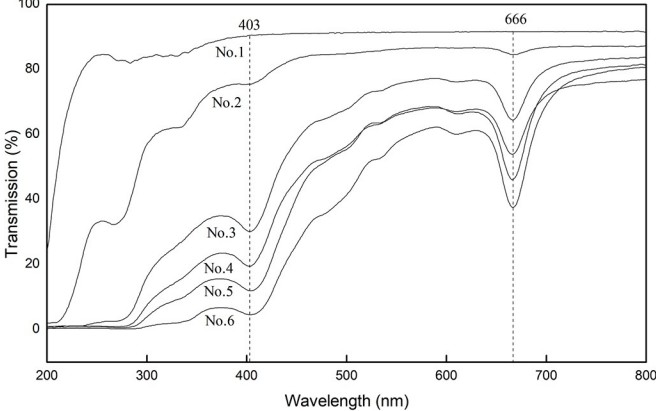
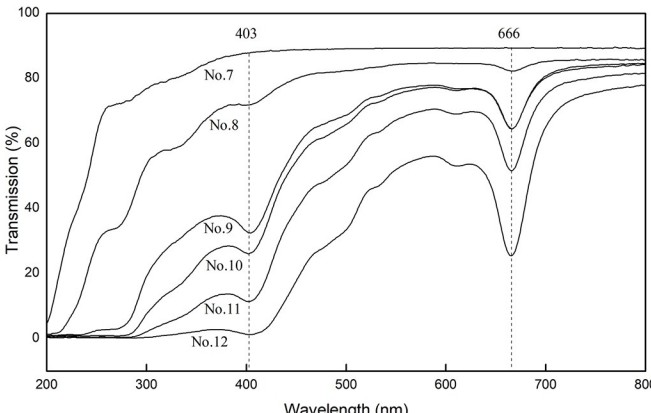

**Fig 3. Transmittance of the PVA-based films at 200–800 nm.**

**Table 4. Transmission (%) the of the PVA-based films at two critical wavelengths[c].**

| Film No. | 1 | 2 | 3 | 4 | 5 | 6 | 7 | 8 | 9 | 10 | 11 | 12 |
|---|---|---|---|---|---|---|---|---|---|---|---|---|
| in 403 nm | 90.5 | 75.5 | 30.2 | 19.5 | 12.0 | 4.6 | 87.8 | 71.9 | 32.6 | 26.2 | 11.4 | 1.3 |
| in 666 nm | 91.7 | 84.7 | 64.5 | 53.9 | 46.1 | 37.6 | 89.4 | 82.3 | 64.6 | 64.6 | 51.7 | 25.5 |

[c] The data in the table is the data of a test.

important in the field of food packaging, which can block the effect of ultraviolet and visible light on food, such as fresh-cut broccoli, soliddrinks, etc., to increase their life.

## Barrier properties of the PVA-based films to deionized water

**Water resistance and swelling ratio.** The water resistance and swelling ratio with the surface hydrophobicity can show the barrier properties of the PVA-based films to deionised water. Fig 4 shows the water resistance and swelling ratio of different PVA-based films. The swelling ratio is an important index to characterize the water resistance of a material, and the smaller the swelling ratio, the better the water resistance for a material. The swelling ratio of the PVA film and PVA/CNF film are both higher than 100%, which is due to the fact that they not only adsorb water on the surface, but also can form hydrogen bond with water with a large number of hydrophilic hydroxyl groups. Compared with the PVA and the PVA/CNF films, the swelling ratio of the CY/PVA and CY/CNF/PVA films (No.3-6 and No.9-12) is almost reduced by half, indicating that the water resistance of them is remarkably enhanced. Moreover, the swelling ratio decreases slowly with the increase of cyanobacteria content in the composite. This is attributed to two aspects. One is that the active amine in guanidine hydrochloride reacts with lipids in cyanobacteria and PVA, as well as with hydroxyl groups in starch, thereby reducing the amount of hydrophilic hydroxyl in the composite film. Second, the cyanobacteria cell wall has a limiting and blocking effect on the entry of water molecules into the complex membrane [36].

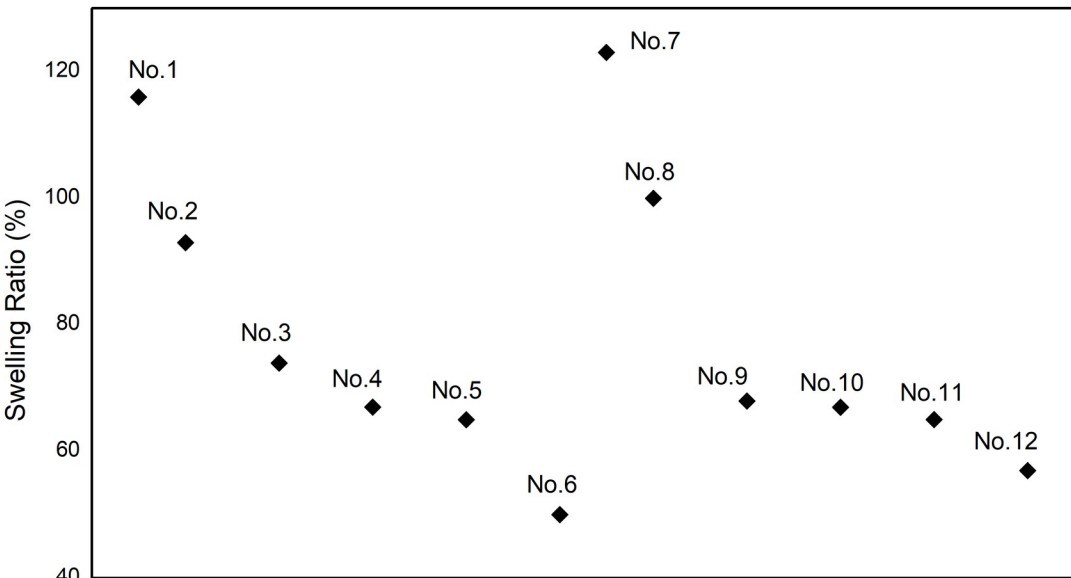

**Fig 4. Water absorption and thickness swelling of different PVA-based films.**

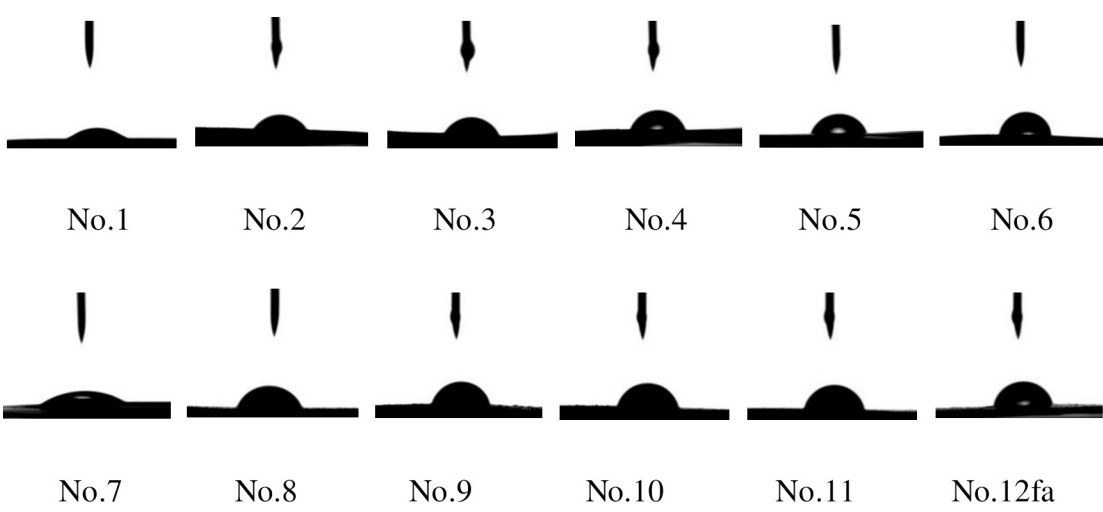

**Fig 5. Contact angles of the prepared PVA-based films to deionized water.**

**Surface hydrophobicity of the PVA-based films.** The contact angle is usually used to evaluate the surface hydrophobicity of a material. The contact angles of the prepared PVA-based films are shown in Fig 5 and Table 5. Compared to the control (PVA film and PVA/CNF film), the CY/PVA and CY/CNF/PVA films present higher contact angles. Furthermore, a small increase in modified cyanobacteria content leads to a significant increase in the contact angle of the composite. When the modified cyanobacteria content increases from 0 wt% to 5 wt%, 15 wt%, 25 wt%, 35 wt% and 45 wt%, the contact angle of the CY/PVA film increases from 12˚ to 55˚, 62˚, 69˚, 72˚ and 82˚, respectively; and that of the CY/PVA/CNF film ascends from 23˚ to 67˚, 74˚, 75˚, 76˚ and 77˚, respectively. This phenomenon is attributed to the higher surface roughness as a result of introducing modified cyanobacteria into the matrix, which can be observed from the scanning pictures of the PVA-based films in Fig 1.

## Morphology of the PVA-based films

The SEM images of the PVA film (No.1-a), CY/PVA films (No.3-b and No.5-c), CNF/PVA film (No.7-d), and CY/CNF/PVA films (No.9-e and No.11-f) are shown in Fig 6. It can be seen that the PVA film and CNF/PVA film are relatively smooth. After adding modified cyanobacteria, the cross-sections of the CY/PVA films and CY/CNF/PVA films become rougher and unevener. This phenomenon can be used to explain our founding that the contact angle of the composite film goes up with the increase of the modified cyanobacteria content. In addition, as shown in Fig 6(B), 6(C) and 6(E), the modified cyanobacteria is wrapped in the PVA matrix or the CNF/PVA matrix, and the interface is blurred, indicating good compatibility of the modified cyanobacteria with the matrix. However, in Fig 6(F), the modified cyanobacteria is phase-separated from the CNF/PVA matrix, showing that too much cyanobacteria leads to poor compatibility with the matrix. The observed phenomenon that the internal structure of

**Table 5. Contact angles to deionized water of the prepared PVA-based films[d].**

| Film No. | 1 | 2 | 3 | 4 | 5 | 6 | 7 | 8 | 9 | 10 | 11 | 12 |
|---|---|---|---|---|---|---|---|---|---|---|---|---|
| Contact angle/˚ | 12 | 55 | 62 | 69 | 72 | 82 | 23 | 67 | 74 | 75 | 76 | 77 |

[d] The data in the table is the data of a test.

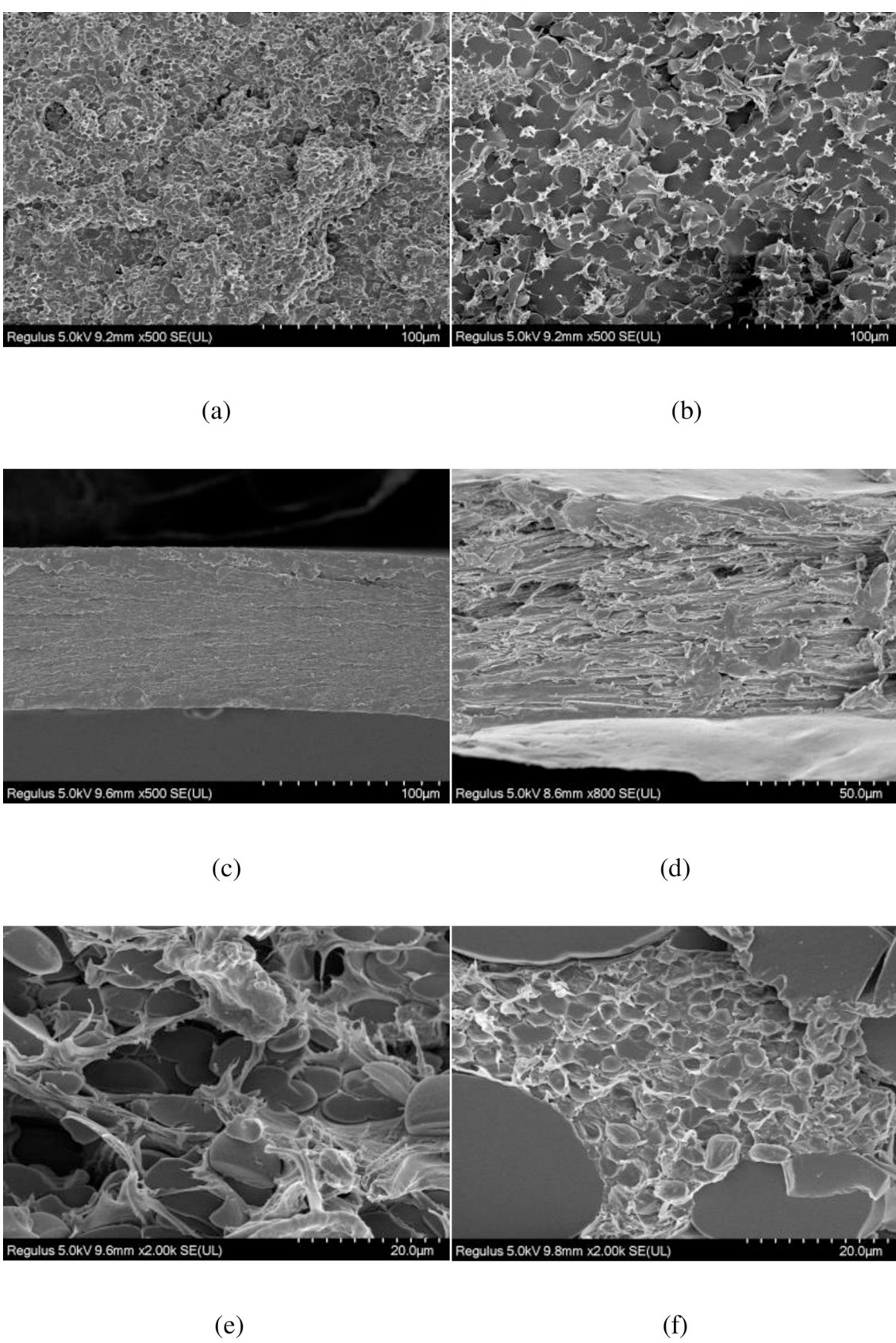

(a)

(b)

(c)

(d)

(e)

(f)

**Fig 6.** SEM images of the PVA film No.1 (a), the CY/PVA film No.3 (b) and No.5 (c), the CNF/PVA film No.7 (d), and the CY/CNF/PVA film No.9 (e) and No.11 (f).

**Table 6. Oxygen transmission rate of the PVA-based films.**

| Film No. | 7 | 8 | 9 | 10 |
|---|---|---|---|---|
| ($cm^3 \cdot cm/cm^3 \cdot s \cdot Pa$) | $1.23 \times 10^{-16}$ | $6.18 \times 10^{-16}$ | $7.38 \times 10^{-16}$ | $3.47 \times 10^{-15}$ |
| | $\pm 9 \times 10^{-18}$ | $\pm 2.5 \times 10^{-17}$ | $\pm 5.7 \times 10^{-17}$ | $\pm 8.2 \times 10^{-16}$ |

the films becomes looser after adding of the cyanobacteria can be used to explain the increased oxygen permeability of the PVA-based film after adding cyanobacteria. It is consistent with the results of our FTIR test.

### Oxygen barrier properties analysis

The oxygen transmission rate of the PVA-based films is shown in Table 6. As the modified cyanobacteria content increases, the oxygen permeability coefficient of the composite presents an ascending trend. It may be due to the looseness and porosity of the inner space of the film after adding cyanobacteria into the matrix, as shown in the SEM images of the PVA-based films in Fig 6. It needs to explore new experimental techniques to improve the oxygen barrier properties in our future research.

### Conclusions

CY/PVA and CY/CNF/PVA films were prepared with a solution casting method. Compared with PVA and CNF/PVA films, the obtained CY/PVA and the CY/CNF/PVA composites exhibit better resistance to light and water. More interestingly, we found that after adding cyanobacteria, the PVA-based films present better barrier properties to blue-violet light and red light. In adddition, introducing cyanobacteria into PVA or CNF/PVA matrix increases the surface roughness and contact angle to water of the composites. This work explored a pathway for utilizing cyanobacteria as raw materials to prepare CY/CNF/PVA films with better resistance to light and water than PVA film.

### Supporting information

**S1 Data.**
(ZIP)

### Acknowledgments

We would like to thank the consulting from Jiangsu Provincial Key Laboratory of Environmental Engineering, Jiangsu Academy of Environmental Science, and the staff on Analysis and Test Center of Nanjing Forestry University for their technical assistance in this project.

### Author Contributions

**Conceptualization:** Sainan Chen.

**Data curation:** Di Liu, Min Qian.

**Formal analysis:** Sainan Chen.

**Funding acquisition:** Sainan Chen.

**Investigation:** Di Liu, Min Qian.

**Methodology:** Sainan Chen, Min Qian, Changyan Xu.

**Project administration:** Sainan Chen, Haiyun Zhou, Jian Bao.

**Resources:** Sainan Chen.

**Supervision:** Changyan Xu.

**Writing – original draft:** Sainan Chen, Di Liu, Min Qian.

**Writing – review & editing:** Sainan Chen, Di Liu, Min Qian, Li Xu, Ying Li, Haozhong Sun, Xi Wang, Haiyun Zhou, Jian Bao, Changyan Xu.

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
