## [Decision Letter · Decision Letter 0]

30 Oct 2019

PONE-D-19-24904

Preparation of cyanobacteria-enhanced poly(vinyl)alcohol-based films with resistance to blue-violet light / red light and water

PLOS ONE

Dear Mrs. Xu,

Thank you for submitting your manuscript to PLOS ONE. After careful consideration, we feel that it has merit but does not fully meet PLOS ONE’s publication criteria as it currently stands. Therefore, we invite you to submit a revised version of the manuscript that addresses the points raised during the review process.

We would appreciate receiving your revised manuscript by Dec 14 2019 11:59PM. To enhance the reproducibility of your results, we recommend that if applicable you deposit your laboratory protocols in protocols.io, where a protocol can be assigned its own identifier (DOI) such that it can be cited independently in the future. For instructions see: http://journals.plos.org/plosone/s/submission-guidelines#loc-laboratory-protocols

We look forward to receiving your revised manuscript.

Kind regards,

Antonio José Félix Carvalho, Ph.D.

Academic Editor

PLOS ONE

Journal Requirements:

3. i) Please provide further details of how the cyanobacteria field sampling took place. Please clarify whether sampling was conducted specifically for this study, and by whom. Please clarify what 'orders' of cyanobacteria refers to. Was any attempt made to characterise and identify the cyanobacteria in the samples? Please provide the geographical coordinates for the sample site. Please also explain in the Methods section whether or not a field permit was obtained, and if not, a brief statement explaining why.

ii) Our internal editors have looked over your manuscript and determined that it may be within the scope of our Plastics in the Environment Call for Papers. The Collection will encompass a diverse range of research articles to better understand various aspects of the effect of plastics in the environment. Additional information can be found on our announcement page: https://collections.plos.org/s/plastics-environment. If you would like your manuscript to be considered for this collection, please let us know in your cover letter and we will ensure that your paper is treated as if you were responding to this call. If you would prefer to remove your manuscript from collection consideration, please specify this in the cover letter.

Additional Editor Comments (if provided):

Please follow the reviewers' instructions and resubmit the manuscript.

Reviewers' comments:

Reviewer's Responses to Questions

**Comments to the Author**

1. Is the manuscript technically sound, and do the data support the conclusions?

Reviewer #1: Yes

Reviewer #2: Partly

2. Has the statistical analysis been performed appropriately and rigorously? 

Reviewer #1: N/A

Reviewer #2: No

3. Have the authors made all data underlying the findings in their manuscript fully available?

Reviewer #1: Yes

Reviewer #2: Yes

4. Is the manuscript presented in an intelligible fashion and written in standard English?

Reviewer #1: Yes

Reviewer #2: No

5. Review Comments to the Author

Reviewer #1: PLOS ONE

Title: Preparation of cyanobacteria-enhanced poly(vinyl)alcohol-based films with resistance to blue-violet light / red light and water

The manuscript describes the preparation of composites based on cyanobacteria incorporated into poly(vinyl)alcohol and cellulose nanofibers/poly(vinyl)alcohol matrices. The scope is interesting once the authors suggest the use of the environmentally harmful cyanobacteria as to generate useful materials. The results and discussion are clear and well described. Some suggestions are listed below in order to improve the reading and reproducibility of some important points.

The title suggested the resistance to blue-violet light / red light and water are important properties, however it was not discussed in the text. I strongly suggest to include the importance of these features. It can be included in the introduction, or in results and discussion.

Line 69 – “In our early elemental analysis test, the contents of carbon, nitrogen and hydrogen in the cyanobacteria sample were 45.02 %, 5.98 % and 9.11 %, respectively.”. Do the authors have already published these data? If positive, please include the reference. If not, please discuss in the results section why it is important.

Table 2- It is need to include a column with the weight content (wt %) of each component of the composite to facilitate the understanding of the results.

Line 134- What is the meaning of CM?

Lines 281-282- please convert the mass values em wt % values, in agreement with the new suggested wt % column from table 2.

Please include a short text correlating the water resistance and swelling ratio with the surface hydrophobicity of the PVA-based films.

How difficult is to collect the cyanobacteria from the sea water? Could be interesting to insert a short text explaining as it could be feasible.

Typing errors:

line 67 -utilizerecycle

131- bsed

line 250 – wto

304 - filmgoes

Reviewer #2: In the present manuscript, the authors described the preparation of poly(vinyl alcohol) (PVA)/nanocellulose composites incorporating cyanobacteria in order to give the materials better light and water resistance. The study is of interest, but some modifications are needed before considering its publication in PLOS One.

- The writing must be reviewed, as some mistakes are present (p. 3, l. 57: “…utilizerecycle cyanobacteria…”; p. 3, l. 62: “The other method of resource utilizing cyanobacteria…”; p. 5, l. 87: “Polyethylene glycol (molecular weight, 200) were purchased…”, among many others).

- The introduction mentions some studies on the preparation of PVA films, but the utilization of cyanobacteria in the context of materials development is neglected. It is important to discuss strategies that were previously reported. In case there is nothing in the literature, the originality must be emphasized.

- Materials and methods: p. 5, l. 86: “After screening, 100 orders of cyanobacteria powder…”. What kind of screening was conducted? What does 100 orders mean? The text must be clear enough to be reproduced by other researchers. Table 2 is not clear. What does CM, CY, CNF and PVA mean? What about the other compounds used in the films formulation (poly(ethylene glycol), guanidine hydrochloride, and carboxymethyl cellulose)? In fact, the reason why these compounds were added, and the amounts chosen must also be explained. The authors conducted water resistance tests, based on the statement: “The swelling of a sample is usually used to indicate the water resistance of a material.”. However, there is no ASTM or any kind of reference to support it. PVA is soluble in water, and therefore the concept of swelling is not correct in this case.

- Results: Table 3 includes results of moisture content, but the methodology for this test was not described in the appropriate section. FTIR of isolated and dried cyanobacteria must be included. The values present in Table 4, Table 5 and Table 6 come from only one measure of each sample? It is important to do the test in triplicate and to present the standard deviation. Why oxygen permeability was conducted only for samples 7-10?

- Reference list is not standardized.

6. PLOS authors have the option to publish the peer review history of their article (what does this mean?). If published, this will include your full peer review and any attached files.

Reviewer #1: No

Reviewer #2: No

---

## [Author Response · Author response to Decision Letter 0]

7 Jan 2020

PONE-D-19-24904

Preparation of cyanobacteria-enhanced poly(vinyl)alcohol-based films with resistance to blue-violet light/red light and water

PLOS ONE

Response to Reviewers

Dear Editors and Reviewers,

Thank you for the time and effort you have spent reviewing our paper. We are pleased to note that you have found our research work interesting and also pointed out some problems to help us improve the quality of our work.

Motivated by your comments, we have deeply reconsidered the architecture of our work and tried to fix all the problems you mentioned. In particular, the revised manuscript of our resubmitted letter has significantly been improved mainly as follows:

Improvement-1: We have modified our manuscript format based on the PLOS ONE style templates. 

Improvement-2: i) We obtained a cyanobacteria field permit and commissioned the staff of the algal and water separation station to perform the field sampling for this study. The station is located on the western bank of Taihu lake. We provided further details of the cyanobacteria field sampling in line 86-100. We characterised and identified the FTIR spectra of cyanobacteria samples in Fig 2. 

ii) We would like our manuscript to be considered for the Plastics in the Environment Call for Papers, as specified in the cover letter. 

Improvement-3: Because all data and other information is already presented in the tset, there is no supporting information section.

Improvement-4: We have revised the manuscript based on the reviewers’ comments (repeated below in italics for your convenience). Our response to each point is as follows:

Reviewer #1: PLOS ONE

Title: Preparation of cynaobacteria-enhanced poly(vinyl)alcohol-based films with resistance to blue-violet light/red light and water

The manuscript describes the preparation of composites based on cyanobacteria incorporated into poly(vinyl)alcohol and cellulose nanofibers/poly(vinyl)alcohol matrices. The scope is interesting once the authors suggest the use of the environmentally harmful cyanobacteria as to generate useful materials. The results and discussion are clear and well described. Some suggestions are listed below in order to improve the reading and reproducibility of some important points.

The title suggested the resistance to blue-violet light / red light and water are important properties, however it was not discussed in the text. I strongly suggest to include the importance of these features. It can be included in the introduction, or in results and discussion. 

Our response: 

Thank you for your nice suggestion. We added the discussion of these important properties in line 264-267.

Line 69 – “In our early elemental analysis test, the contents of carbon, nitrogen and hydrogen in the cyanobacteria sample were 45.0 2%, 5.98 % and 9.11 %, respectively.”. Do the authors have already published these data? If positive, please include the reference. If not, please discuss in the results section why it is important.

Our response: 

These data have not been published before, which came from the extra tests we did in the research. They were put into the introduction to provide more information about cyanobacteria. We decided to delete these data after careful conideration to avoid unnecessary misunderstandings.

Table 2- It is need to include a column with the weight content (wt %) of each component of the composite to facilitate the understanding of the results.

Our response:

The weight content (wt%) of the modified cyanobacteria (CM) was added to help understand the results in Table 2. The wt% of CM in film No.1-6 is CM to poly(vinyl)alcohol. The wt% of CM in film No.7-12 is CM to cellulose nanofibers/poly(vinyl)alcohol. 

Line 134- What is the meaning of CM?

Our response:

Thank you for pointing out our negligence. The meaning of CM is the modified cyanobacteria, and a description was added to line 136. 

Lines 281-282- please convert the mass values em wt % values, in agreement with the new suggested wt % column from table 2.

Our response: 

Thanks for your suggestion. We have modified it.

Please include a short text correlating the water resistance and swelling ratio with the surface hydrophobicity of the PVA-based films.

Our response:

Thank you for pointing out the problem. We added a short text to illustrate this in line 278-279.

How difficult is to collect the cyanobacteria from the sea water? Could be interesting to insert a short text explaining as it could be feasible.

Our response:

Thanks for your suggestion. We inserted a short text explaining of collecting cyanobacteria in line 90-100.

Typing errors:

line 67 -utilizerecycle

131- bsed

line 250 – wto

304 - filmgoes

Our response:

Thank you for your careful attention to our typing errors. We have corrected these.

Reviewer #2: In the present manuscript, the authors described the preparation of poly(vinyl alcohol) (PVA)/nanocellulose composites incorporating cyanobacteria in order to give the materials better light and water resistance. The study is of interest, but some modifications are needed before considering its publication in PLOS One.

- The writing must be reviewed, as some mistakes are present (p. 3, l. 57: “…utilizerecycle cyanobacteria…”; p. 3, l. 62: “The other method of resource utilizing cyanobacteria…”; p. 5, l. 87: “Polyethylene glycol (molecular weight, 200) were purchased…”, among many others).

Our response:

Thank you for your careful attention to our writing mistakes. We have corrected these.

- The introduction mentions some studies on the preparation of PVA films, but the utilization of cyanobacteria in the context of materials development is neglected. It is important to discuss strategies that were previously reported. In case there is nothing in the literature, the originality must be emphasized.

Our response:

Thank you for your nice suggestion. We added thedescriotion of the originality of cyanobacteria used directly in the preparation of composite films. 

- Materials and methods: p. 5, l. 86: “After screening, 100 orders of cyanobacteria powder…”. What kind of screening was conducted? What does 100 orders mean? The text must be clear enough to be reproduced by other researchers. 

Our response:

Thank you for pointing out the problem. We screened cyanobacteria for particle size. “order” is a writing mistake and has been modified to “mesh”.

Table 2 is not clear. What does CM, CY, CNF and PVA mean? 

Our response:

Thank you for pointing out the problem. We have modified Table 2. CM, CY, CNF and PVA are abbreviations of modified cyanobacteria, cyanobacteria, cellulose nanofibers and poly(vinyl)alcohol. Their meanings are mentioned in line 136, 51, 47 and 33, respectively.

What about the other compounds used in the films formulation (poly(ethylene glycol), guanidine hydrochloride, and carboxymethyl cellulose)? In fact, the reason why these compounds were added, and the amounts chosen must also be explained. 

Our response:

The reasons why these compounds are putted in line 76-80, and the explanation for the amounts chosen is putted in line 131-133.

The authors conducted water resistance tests, based on the statement: “The swelling of a sample is usually used to indicate the water resistance of a material.”. However, there is no ASTM or any kind of reference to support it. PVA is soluble in water, and therefore the concept of swelling is not correct in this case.

Our response:

Thank you for pointing out the problem. We added an ASTM standard to support it in line 180.

- Results: Table 3 includes results of moisture content, but the methodology for this test was not described in the appropriate section. 

Our response:

The methodology for this test was added in line 180-184.

FTIR of isolated and dried cyanobacteria must be included. 

Our response:

FTIR of isolated and dried cyanobacteria is labeled “CY” in Fig 2, 

The values present in Table 4, Table 5 and Table 6 come from only one measure of each sample? It is important to do the test in triplicate and to present the standard deviation. 

Our response:

i)Thank you for your suggestion. The standard deviation have been added in Table 6.

ii) The transmission test is to show that the poly(vinyl)alcohol-based films greatly enhances the barrier properties to blue-violet light/red light after mixing cyanobacteria in Table 4. The contact angle test is to show that the poly(vinyl)alcohol-based films greatly enhances the surface hydrophobicity in Table 5. Because the results of these two tests are special effective and do not pursue overly detailed values, only one test was done.

Why oxygen permeability was conducted only for samples 7-10?

Our response:

We screened representative samples for oxygen permeability testing based on previous test results. In addition, because the oxygen permeability of samples 7-10 is regular, other samples were not tested.

- Reference list is not standardized.

Our response:

Thank you for your careful attention to our manuscript. We have corrected these.

We hope this paper is suitable for PLOS ONE and it can help a little bit for people studying similar fields. 

We deeply appreciate your consideration of our manuscript. If there are any other modifications we could make, we would like very much to modify them. We look forward to receiving comments from the reviewers. If you have any queries, please don’t hesitate to contact me at the address below.

Thank you and best regards.

Yours sincerely,

Changyan Xu

Corresponding author:

Name: Changyan Xu, Sainan Chen

E-mail: changyanxu1999@163.com, csn@jshb.gov.cn

---

## [Editor Report · Decision Letter 1]

24 Jan 2020

Preparation of cyanobacteria-enhanced poly(vinyl)alcohol-based films with resistance to blue-violet light / red light and water

PONE-D-19-24904R1

Dear Dr. Xu,

We are pleased to inform you that your manuscript has been judged scientifically suitable for publication and will be formally accepted for publication once it complies with all outstanding technical requirements.

With kind regards,

Antonio José Félix Carvalho, Ph.D.

Academic Editor

PLOS ONE

---

## [Editor Report · Acceptance letter]

31 Jan 2020

PONE-D-19-24904R1 

Preparation of cyanobacteria-enhanced poly(vinyl)alcohol-based films with resistance to blue-violet light / red light and water 

Dear Dr. Xu:

I am pleased to inform you that your manuscript has been deemed suitable for publication in PLOS ONE. Congratulations! Your manuscript is now with our production department. 

With kind regards,

on behalf of

Dr. Antonio José Félix Carvalho 

Academic Editor

PLOS ONE